# Clinical Validation of a Urine Test (Uromonitor-V2^®^) for the Surveillance of Non-Muscle-Invasive Bladder Cancer Patients

**DOI:** 10.3390/diagnostics10100745

**Published:** 2020-09-24

**Authors:** Caroline A. Sieverink, Rui P. M. Batista, Hugo J. M. Prazeres, João Vinagre, Cristina Sampaio, Ricardo R. Leão, Valdemar Máximo, J. Alfred Witjes, Paula Soares

**Affiliations:** 1Department of Urology, Radboud University Nijmegen Medical Center, Geert Grooteplein Zuid 10, 6525 GA Nijmegen, The Netherlands; Fred.Witjes@radboudumc.nl; 2i3S-Instituto de Investigação e Inovação em Saúde, R. Alfredo Allen 208, 4200-135 Porto; Portugal; rbatista@ipatimup.pt (R.P.M.B.); hprazeres@i3s.up.pt (H.J.M.P.); jvinagre@ipatimup.pt (J.V.); sampaio.csm@gmail.com (C.S.); vmaximo@ipatimup.pt (V.M.); psoares@ipatimup.pt (P.S.); 3Instituto de Patologia e Imunologia Molecular da Universidade do Porto (IPATIMUP), Rua Júlio Amaral de Carvalho 45, 4200-135 Porto, Portugal; 4U-Monitor Lda, Rua Alfredo Allen, Nº 461 Paranhos, 4200-461 Porto, Portugal; 5Department of Pathology, Faculdade de Medicina, Universidade do Porto, Alameda Prof. Hernâni Monteiro, 4200-319 Porto, Portugal; 6Department of Urology, Hospital de Braga, Sete Fontes—São Victor, 4710-243 Braga, Portugal; romaoleao@gmail.com; 7Department of Urology, Hospital CUF Coimbra, Rua Camilo Pessanha 1, 3000-600 Coimbra, Portugal; 8Faculdade de Medicina de Coimbra (FMUC), Rua Larga 2, 3000-370 Coimbra, Portugal

**Keywords:** biomarkers, bladder cancer, cystoscopy, cytology, follow-up, non-muscle-invasive bladder cancer, recurrence, urine test

## Abstract

The costly and burdensome nature of the current follow-up methods in non-muscle-invasive bladder cancer (NMIBC) drives the development of new methods that may alternate with regular cystoscopy and urine cytology. The Uromonitor-V2^®^ is a new urine-based assay in the detection of hotspot mutations in three genes (*TERT*, *FGFR3*, and *KRAS*) for evaluation of disease recurrence. The aim of this study was to investigate the Uromonitor-V2^®^’s performance in detecting NMIBC recurrence and compare it with urine cytology. From February 2018 to September 2019 patients were enrolled. All subjects underwent a standard-of-care (SOC) cystoscopy, either as part of their follow-up for NMIBC or for a nonmalignant urological pathology. Urine cytology was performed in NMIBC patients. Out of the 105 patients enrolled, 97 were eligible for the study. Twenty patients presented nonmalignant lesions, 29 had a history of NMIBC with disease recurrence, and 49 had a history of NMIBC without recurrence. In NMIBC, the Uromonitor-V2^®^ displayed a sensitivity, specificity, positive predictive value (PPV), and negative predictive value (NPV) of 93.1%, 85.4%, 79.4%, and 95.3%, respectively. Urine cytology was available for 52 patients, and the sensitivity, specificity, PPV, and NPV were 26.3%, 90.9%, 62.5%, and 68.2%, respectively. With its high NPV of 95.3%, the Uromonitor-V2^®^ revealed promising properties for the follow-up of patients with NMIBC.

## 1. Introduction

Bladder cancer (BCa) is the tenth most common type of cancer worldwide with more than 500,000 newly diagnosed cases in 2018 [1]. The disease can be stratified in two subtypes: non-muscle-invasive bladder cancer (NMIBC) and muscle-invasive bladder cancer (MIBC). At initial diagnosis, approximately 75% of the patients present with NMIBC [2]; the majority of these patients will develop disease recurrence. Less frequent, although more ominous, is the development of muscle-invasive disease by patients previously diagnosed with NMIBC. Disease progression in NMIBC occurs in up to 20% of the patients, with carcinoma in situ (CIS) and/or T1 high-grade (HG) patients being most at risk [3,4,5,6,7].

The high rates of disease recurrence and the risk for disease progression determine that frequent follow-up regime is highly required. The current follow-up schedule of patients with a history of NMIBC consists of regular surveillance cystoscopies in combination with urine cytology. Follow-up through regular cystoscopies and cytology should be maintained years following diagnosis and can even span throughout life [8]. These extensive schemes lead to high costs, making BCa the most expensive cancer when considering patients’ lifetime expenses [9]. Another drawback of the current follow-up is the invasive nature of the cystoscopic procedure and the low sensitivity of urine cytology, particularly in low-grade NMIBC [10]. There is a need for cheaper, better, and less invasive follow-up methods in NMIBC patients. One potential way to improve the current regime is through the introduction of urinary biomarkers, either as an addition to current standard practice or as a (partial) replacement of the current methods. The properties required for adequate testing vary and depend on the clinical goal. For biomarker-based tests that aim to (partially) replace cystoscopies in the follow-up of NMIBC, a high negative predictive value (NPV) should be mandatory. A high NPV will assure the urologist that no tumors are left undiagnosed. Over the last decades, a multitude of biomarker-based urine tests were developed and were granted FDA approval. Tests like the NMP22 BladderCheck, UroVysion, or BTA stat failed to be implemented into clinical practice due to insufficient NPV and low specificity.

In recent years, several biomarker-based tests with significantly higher NPV emerged. Although promising, these tests need to be validated in a clinical setting before implementation [11]. Recently, a new urine test (the Uromonitor-V2^®^, U-monitor, Porto, Portugal) was presented. The test evaluates a subset of hotspot alterations in three different genes (*TERT*, *FGFR3*, and *KRAS*) using real-time qPCR. These targeted alterations include some of the most common genetic events in NMIBC. Oncogene-activating mutations in fibroblast growth factor receptor 3 (*FGFR3*) are one of the most relevant drivers of urothelial transformation. These mutations are reported in BCa with an overall frequency of ~35% [12,13] and account for 80% of the early stage and low-grade tumors; although it presents potential for accurate detection of low-grade tumors, it fails significantly to provide acceptable sensitivity in the detection of high-grade tumors [14,15,16,17,18]. Mutations in *RAS* genes are found in a lower percentage (13%) of BCa. This frequency should be taken into account when finding new solutions in NMIBC follow-up, in particular since *KRAS* and *FGFR3* mutations are mutually exclusive events [19]. *TERT* promoter (*TERTp*) mutations are reported with different frequencies in several cancers, but particularly in NMIBC are present in up to 80% across different stages/grades and are absent in inflammatory or urinary infections [20,21,22,23]. *TERTp’s* transversal high frequencies across different stages and grades elects it as a high-value “all-round” biomarker to be used in clinical practice of NMIBC detection [24,25].

There is currently a gap in the literature regarding the true prevalence of these alterations in NMIBC recurrences and the sensitivity in the detection of recurrent lesions when screened in combination. Since these alterations present, at this time, some of the most common genetic events in NMIBC tumors, it became imperative to elucidate their potential to detect NMIBC recurrence in urine. In a previous multicenter study on NMIBC patients, in follow-up the test showed promising results with a sensitivity, specificity, and NPV of respectively 100%, 83.8%, and 100% in a subgroup of 24 NMIBC recurrent patients [26]. With this current study, we aimed to validate these results in an independent cohort of NMIBC patients in follow-up.

## 2. Results

### 2.1. Patients’ Characteristics

From the 105 enrolled subjects, eight were deemed ineligible for inclusion, leading to 97 patients being eligible for inclusion in the study. Five of the ineligible patients were previously diagnosed with a muscle-invasive tumor; one failed Uromonitor-V2^®^ detection, whereas the remaining two samples did not have sufficient material for testing. Of the 97 eligible patients, 77 presented a history of NMIBC. Of these, 29 were positive for recurrence during enrollment cystoscopy. An overview of the clinicopathological information of the eligible patients is presented in Table 1. The three subject groups had no significantly different characteristics, except for the time since last treatment between the NMIBC patients with and without current recurrence. The time since last treatment was significantly longer in patients with recurrence (10.81 months, SD ± 18.84) in comparison with those without recurrence (5.20 months, SD ± 6.31) (*p* = 0.004).

### 2.2. Test Results

Out of the 97 patients who were successfully tested with Uromonitor-V2^®^, 36 (37.1%) patients presented a positive result, while the remaining 61 (62.9%) were negative. Of the 97 patients, 29 subjects were positive for disease recurrence at time of enrollment. The Uromonitor-V2^®^ was able to identify 27 of the recurrent samples; two samples were undetected. Histological material from transurethral resection of the bladder tumor (TURBT), or biopsy of the tumor was available in fourteen of the recurrence-positive patients. Of the pathologically proven recurrences, six were diagnosed with a Ta tumor, two with a T1 tumor, five patients with CIS, and one patient progressed to a muscle-invasive tumor. All fourteen of the histologically proven patients had a positive result in the Uromonitor-V2^®^ test. The fifteen recurrences without histological evidence were clinically diagnosed as low-grade Ta tumors by the attending urologist based on tumor history and tumor appearance.

In total, 36 samples were Uromonitor-V2^®^-positive, of which 27 were recurrence-positive. For the remaining positive samples, seven were of patients with a history of NMIBC without a detectable recurrence, while two were of patients without a history of bladder cancer. Follow-up data were available for six of the Uromonitor-V2^®^ false-positive NMIBC patients. At six months follow-up, five patients had no sign of disease recurrence and one patient was suspected for CIS. No pathological confirmation for the suspected CIS was obtained due to the patient’s poor clinical condition. In the two benign patients with a positive Uromonitor-V2^®^ result, one patient underwent a cystoscopy for lower urinary tract symptoms due to benign prostate hyperplasia, while the other patient presented with urge incontinence.

The Uromonitor-V2^®^ presented an overall sensitivity of 93.1% (27/29), a specificity of 86.8% (59/68), a positive predictive value (PPV) of 75.0% (27/36), and an NPV of 96.7% (59/61). In Table 2, an overview is presented comparing the recurrence-positive patients versus nonrecurrence patients (control patients + non-recurrent NMIBC patients). When only including patients with a history of NMIBC in the analysis, and excluding the “healthy” control patients, the test characteristics remained comparable with a sensitivity of 93.1% (27/29), a specificity of 85.4% (41/48), a PPV of 79.4% (27/34), and an NPV of 95.3% (41/43).

### 2.3. Cytology

Results of cytology, collected at enrollment cystoscopy, were available for 52 samples. Nineteen were from patients with recurrent disease, of which five had a positive cytology (TPS4/TPS5); one cytology result was equivocal (TPS3); while the remaining were negative (TPS2). All patients who had positive cytologies were positive for the Uromonitor-V2^®^. In this study, urine cytology showed a sensitivity, a specificity, a PPV, and an NPV of 26.3%, 90.9%, 62.5%, and 68.2%, respectively. An overview of the Uromonitor-V2^®^ test characteristics in comparison with cytology is presented in Table 3. In Figure 1, Receiver Operating Characteristic (ROC) curves of both urine cytology and the Uromonitor-V2 for patients with a history of NMIBC are presented, showing an Area Under The Curve (AUC) of 0.586 and 0.893, respectively.

## 3. Discussion

The high number of cases with disease recurrence in NMIBC patients can be partially explained by inadequacies of the current follow-up methods. While white light cystoscopy (WLC) is an adequate method for detecting papillary lesions, it lacks adequacy in the detection of, for example, CIS or microscopic lesions [3]. Methods to improve tumor visualization, like blue light cystoscopy (BLC) and narrow-band imaging (NBI), were, therefore, developed. In BLC, a fluorescent “dye” is accumulated in neoplastic cells, while in NBI the red spectrum of white light enhances the visual differentiation between hypovascularized BCa tissue and normal tissue counterpart. These methods were promising in more accurate monitoring of disease recurrence but are not widely used [8,27,28,29]. The main advantage of urine cytology is the noninvasive nature of the procedure. Urine cytology has a high specificity (86%) but low sensitivity, especially in patients with low-grade disease. The sensitivity of urine cytology ranges from 16% in low-grade patients up to 84% in patients in high-grade disease and is only recommended as a complement to cystoscopy in patients with high-grade disease [8,10]. Additionally, cytology is flawed by wide interobserver and intraobserver variability, and reporting results can take up to a few days. In Table 4 is presented a summarized overview of the performance and test comparison of urine cytology and the Uromonitor-V2^®^. The current pitfalls of the follow-up methods in BCa result in a high economic burden with BCa being accountable for 3% of all cancer costs within the European Union [30]. These reasons led to the development of new and noninvasive methods of follow-up in NMIBC. The use of Uromonitor-V2^®^ to monitor recurrence in patients with NMIBC revealed a sensitivity, specificity, PPV, and NPV of 93.1%, 85.4%, 79.4%, and 95.3%, respectively. The latter contrasts with urine cytology results and, available for 52 patients, displayed a decrease in sensitivity, specificity, PPV, and NPV (26.3%, 90.9%, 62.5%, and 68.2%, respectively). Over the last decades, multiple biomarker-based urine assays were developed to act as an additive to the current follow-up methods, or even as a (partial) replacement. Tests such as the NMP22, BTA stat, or UroVysion were FDA-approved, but none of them is widely used routine as no relevant NPV is achieved [11]. Some of the latest additions (the CxBladder Monitor, the Xpert Bladder Monitor, and the Bladder EpiCheck) present more promising results. In the overall BCa population, the Xpert Bladder Monitor reports an NPV of 93%, being surpassed by the CxBladder and the Bladder Epicheck displaying an NPV of 95%. The Xpert Bladder Monitor and the Bladder Epicheck perform better (higher NPVs) in high-grade tumors (NPV of 98% and 99.3%, respectively) [31,32,33]. For Uromonitor-V2^®^, an NPV of 95.3% was achieved, demonstrating a high performance in screening the cohort of NMIBC patients. The use of well-established technologies and affordable equipment leads to clear advantages in the implementable of Uromonitor-V2^®^ capacity in different centers and reduced test costs; the entire procedure relies on affordable and easy-to-use components. It is based on the use of real-time qPCR, a methodology that is well implemented in most laboratories, allowing in-house testing. Another clear advantage of the test is the short six-hour period response that creates a binary response, a yes-or-no result. While only being tested in a restricted number of samples, the presence of nonmalignant urological pathologies did not interfere with the Uromonitor-V2^®^ test performance. As a drawback of this study, it is necessary to point out the relatively low number of histologically confirmed recurrences. Histological confirmation was available in fourteen of the 29 patients. Whether a patient with disease recurrence was referred for tumor removal was a decision based on the visual appearance of the tumor and prior tumor history. All the resected bladder tumors were detected by the Uromonitor-V2^®^. Only two tumors detected through cystoscopy and/or cytology, but not confirmed by histology, were unidentified by the Uromonitor-V2^®^; they corresponded to patients previously diagnosed with pTa, low-grade tumors, and in both, the tumors were relatively small (max. diameter 5 mm and 12 mm). These tumors might represent false-positive cystoscopy or cytology results, however without histological confirmation this cannot be demonstrated. Still, missing these small, low-grade tumors could be considered a defendable act since they do not determine the course of the disease, and can be followed in an active surveillance strategy. No sub-analysis on high-grade recurrences was conducted due to the relatively low number of pathologically confirmed recurrences. It would be interesting to test the Uromonitor-V2^®^ in a larger cohort of high-grade NMIBC patients in a future research study. This study would also benefit from longer-term follow-up data, especially in NMIBC patients who presented a positive Uromonitor-V2^®^ test but were considered recurrence-negative, according to current follow-up methods. These false-positive patients are of extreme interest since they might harbor microscopically recurrences that are not detected by the current combination of cystoscopy and cytology, and the Uromonitor-V2^®^ would be an invaluable tool in this screening setting.

## 4. Materials and Methods

### 4.1. Patients

This was a prospective, blinded, single-visit, case-enriched cohort study. All patients in this study were under follow-up at the Radboud University Medical Center in Nijmegen, the Netherlands, either because of a history of NMIBC or because of a benign, non-BCa related urological pathology. Next to enrolling patients with a history of NMIBC, a subgroup of non-BCa patients were enrolled to test whether the presence of nonmalignant urological pathologies would interfere with the Uromonitor-V2^®^ test results. All were enrolled prior to undergoing a cystoscopy at the hospital’s outpatient clinic. Three groups of patients were subsequently identified: (1) patients with a history of NMIBC with a recurrence at time of cystoscopy; (2) patients with a history of NMIBC without a recurrence at cystoscopy; and (3) patients without a history of BCa, undergoing cystoscopy for benign urological causes. Patients were eligible for inclusion if they were ≥18 years of age, able to give written consent and provide a minimum of 10 mL of urine prior to undergoing standard-of-care (SOC) cystoscopy. NMIBC patients in follow-up could be enrolled if they had an event (initial or recurrent NMIBC) within five years prior to enrollment. Exclusion criteria were inadequate material for testing; a failed Uromonitor-V2^®^ test; previous diagnosis with muscle invasive bladder; or, in case of the benign control group, any prior history of bladder cancer.

Clinical information of all patients was collected. Informed consent was obtained from all patients. All procedures described in this study were in accordance with national and institutional ethical standards and the Declaration of Helsinki, and the trial was approved (19 December 2017) by the local ethics committee.

### 4.2. Urine Collection, Sample Handling and Testing

All urine collections were carried out within the standard clinical surveillance program of the participating patients, in which all patients came in for a regular follow-up cystoscopy and were asked to provide additional urine in parallel. Additional cytology was collected for those patients with a history of NMIBC. Following urine collection, all patients underwent an SOC cystoscopy by a urologist or a urologist in training who inspected the bladder for any abnormalities. Recurrence was defined as either pathologically proven disease following TURBT or was the clinical decision of the attending urologist.

After collection, the urine was filtered using a pretreated 0.80 µm nitrocellulose syringe filter (Whatman^®^ Filter-Z612545, Merck, Darmstadt, Germany) containing a house-made conservative storage buffer [26]. Per patient, ≥10 mL of urine was collected. A minimum of two filters per patient was required to perform adequate testing, with a minimum amount of 5 ml being used per filter. After the filtration process, the filters could be stored at 4 °C for at least a month, before being shipped at room temperature to the laboratory (Uromonitor, Porto, Portugal) for further testing. High-molecular-weight DNA was extracted from the filters using the Norgen^®^ Plasma/Serum Cell-Free Circulating DNA Purification Mini Kit (Norgen Biotek Corp, Thorold, ON, Canada) as described [26]. *TERT*, *FGFR3*, and *KRAS* testing was performed on 25–50 ng of the extracted DNA. The extracted DNA was amplified and detected on a qPCR real-time machine (Applied Biosystems QS5, Thermo Fisher Scientific, Waltham, MA, USA) using the proprietary chemistry for amplification and detection, as provided in the Uromonitor-V2^®^ test kit. Amplification signals were analyzed as recommended by the manufacturer (Uromonitor, Porto, Portugal). If at least one of the screened alterations provided a positive result, then the test was positive. Refer to Figure 2 for a visualization of the workflow.

### 4.3. Statistical Analyses

Statistical analysis was carried out using 21.0 SPSS Statistical Package (SPSS, Inc., Chicago, IL, USA, 220 2003). Descriptive statistics were performed and differences between groups were tested by the Student’s *t*-test, Mann–Whitney test, or one-way ANOVA, according to variables and groups.

## 5. Conclusions

With a sensitivity of 93.1%, specificity of 85.4%, and an NPV of 95.3% this study presents Uromonitor-V2^®^ as a promising test with characteristics for detecting recurrence in NMIBC patients under follow-up. The test displayed its potential as an alternative to the current follow-up methods, in the current validation study and in the previous study on the Uromonitor-V2^®^, with comparable findings [26]. Further research in a larger cohort of high-grade NMIBC patients through a phase III multicenter trial should be conducted to determine whether the Uromonitor-V2^®^ could serve as a (partial) replacement for cystoscopy and/or cytology.

## Figures and Tables

**Figure 1 diagnostics-10-00745-f001:**
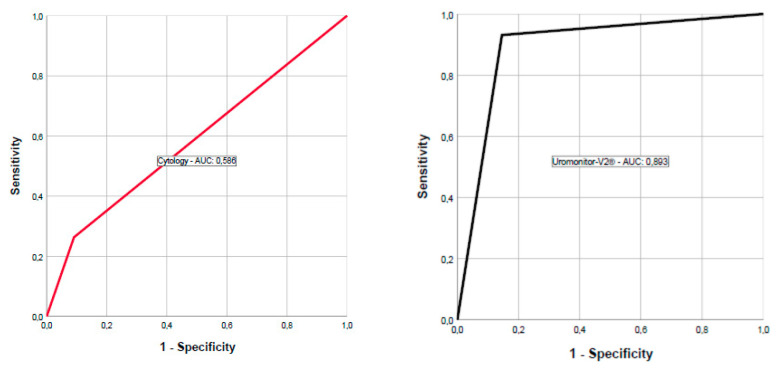
Comparing ROC curves of urine cytology (red) with Uromonitor-V2^®^ (black) for patients with a history of NMIBC.

**Figure 2 diagnostics-10-00745-f002:**
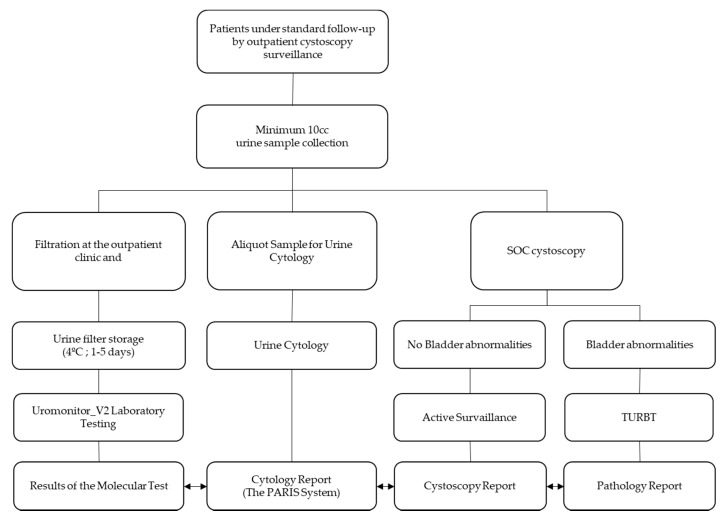
A visualization of the Uromonitor-V2^®^ study workflow. SOC, standard-of-care; TURBT, transurethral resection of the bladder tumor.

**Table 1 diagnostics-10-00745-t001:** Clinicopathological information on all enrolled study subjects (*n* = 97).

Characteristics	Non Bladder Cancer (*n* = 20)	NMIBC Recurrence (*n* = 29)	NMIBC Nonrecurrence (*n* = 48)
Age			
Median (min–max)	71 (22–82)	68 (50–85)	72.5 (49–93)
Gender			
Female	8 (40%)	9 (31%)	12 (25%)
Male	12 (60%)	20 (69%)	36 (75%)
Smoking			
No	1 (5%)	4 (13.8%)	6 (12.5%)
Yes, former	2 (10%)	19 (65.5%)	35 (7.9%)
Yes, current	0 (0%)	5 (17.2%)	7 (14.6%)
Unknown	17 (85%)	1 (3.4%)	0 (0%)
Most recent intravesical treatment			
Chemotherapy	N.A.	13 (44.8%)	21 (43.8%)
BCG	N.A.	5 (17.2%)	11 (22.9%)
Synergo	N.A.	8 (27.6%)	13 (27.1%)
Other	N.A.	1 (3.4%)	1 (2.1%)
None	N.A.	2 (6.9%)	2 (4.2%)
Time since last treatment (in months)			
Mean (min–max)	N.A.	10.81 (1–56)	5.20 (0–28)
Stage initial tumor			
PUNLMP	N.A.	2 (6.9%)	0 (0%)
pTa	N.A.	17 (58.6%)	27 (56.3%)
pT1	N.A.	3 (10.3%)	8 (16.7%)
CIS	N.A.	7 (24.1%)	13 (27.1%)
Grade initial tumor			
Low-grade	N.A.	13 (44.8%)	19 (39.6%)
High-grade	N.A.	16 (55.2%)	29 (60.4%)
Stage last recurrence			
pTa	N.A.	6 (20.7%)	N.A.
pT1	N.A.	2 (6.9%)	N.A.
CIS	N.A.	5 (17.2%)	N.A.
MIBC	N.A.	1 (3.5%)	N.A.
Not available	N.A.	15 (51.7%)	N.A.
Grade last recurrence			
Low-grade	N.A.	2 (6.9%)	N.A.
High-grade	N.A.	12 (41.4%)	N.A.
Not available	N.A.	15 (51.7%)	N.A.
Cytology at enrollment cystoscopy			
TPS2	N.A.	13 (44.8%)	28 (58.3%)
TPS3	N.A.	1 (3.4%)	2 (4.2%)
TPS4	N.A.	3 (10.3%)	2 (4.2%)
TPS5	N.A.	2 (6.9%)	1 (2.1%)
Not available	N.A.	10 (34.5%)	15 (31.3%)

PUNLMP, papillary urothelial neoplasm of low malignant potential; N.A., non-applicable; BCG, Bacillus Calmette-Guérin; TPS, The Paris System for Reporting Urinary Cytology.

**Table 2 diagnostics-10-00745-t002:** Test performances in enrolled patients (*n* = 97), per subgroup.

Test Result	Recurrence-Positive		Recurrence-Negative		Total
	Confirmed by Histology	Confirmed by Cytology/Cystoscopy	Non-Recurrent NMIBC	Non-BCa	
Positive	14	13	7	2	36
Negative	0	2	41	18	61
Total	14	15	48	20	97

**Table 3 diagnostics-10-00745-t003:** Comparing Uromonitor-V2^®^ test characteristics with urine cytology.

Patients	Parameter	*n*/*N*	Results, % (95% CI)
Uromonitor-V2^®^ NMIBC patients	Sensitivity	27/29	93.1 (75.8–98.8)
	Specificity	41/48	85.4 (75.8–93.4)
	PPV	27/34	79.4 (57.5–87.3)
	NPV	41/43	95.3 (87.6–99.4)
Uromonitor-V2^®^ All patients	Sensitivity	27/29	93.1 (75.8–98.8)
	Specificity	59/68	86.8 (71.6–93.5)
	PPV	27/36	75.0 (61.6–90.7)
	NPV	59/61	96.7 (82.9–99.2)
Cytology	Sensitivity	5/19	26.3 (10.1–51.4)
	Specificity	30/33	90.9 (74.5–97.6)
	PPV	5/8	62.5 (25.9–89.8)
	NPV	30/44	68.2 (52.3–80.9)

**Table 4 diagnostics-10-00745-t004:** A summarized description of the Uromonitor-V2^®^ test characteristics and requirements compared with cytology.

	Cytology	Uromonitor-V2
Overall Sensitivity	48% [10]	93%
Overall Specificity	86% [10]	85%
Type of sample required	Urine	Urine
Amount of sample recommended	>30 mL [34]	10 mL
Technical time required from sample to result	Couple of days	6 h

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
