# Peer review of "Clinical Validation of a Urine Test (Uromonitor-V2®) for the Surveillance of Non-Muscle-Invasive Bladder Cancer Patients"

_diagnostics, 2020, doi:10.3390/diagnostics10100745_

Round 1
Reviewer 1 Report
Caroline et al. have done interesting work in NMIBC diagnostics. Though work has good impact but several major and minor comments need to be address.
1) Please look for spell check and few grammatical corrections.
2) Author mentioned the details of specificity and sensitivity in the biomarker in the tabular form. Please show the AUC curve data for some major experiments.
3) Comparing Uromonitor-V2® test characteristics or performance with urine cytology, please add summarized description explaining their amount of sample required, time required for assay, sensitivity, specificity of detection, etc.
4) Please include more basic information's or background about urine-based detection methods for bladder cancer in the introduction section.
Author Response
To the reviewer,
We would also like to thank you for taking the time and effort to review our manuscript, your feedback is much appreciated. We would like to use this letter to address the your comments and hope that by implementing the feedback, we have improved our manuscript.
Point 1: Please look for spell check and few grammatical corrections.
Response: Thank you for your suggestion. Please know that the manuscript underwent an extensive check of the English language, as was also suggested by the second reviewer.
Point 2: Author mentioned the details of specificity and sensitivity in the biomarker in the tabular form. Please show the AUC curve data for some major experiments.
Response: Please find two ROC curves added to the manuscript at the results section, following the section on cytology results within our study (section 2.3). These ROC curves show the details of both urine cytology and the Uromonitor-V2 test on patients with a history of NMIBC.
Point 3: Comparing Uromonitor-V2® test characteristics or performance with urine cytology, please add summarized description explaining their amount of sample required, time required for assay, sensitivity, specificity of detection, etc.
Response: Based on this comment we decided to add a small figure at the end of the discussion section summarizing the results of both cytology in literature and the Uromonitor-V2 test within our study. Please see figure 2.
Point 4: Please include more basic information's or background about urine-based detection methods for bladder cancer in the introduction section.
Response: A more extensive explanation on the history of biomarker-based urine tests was added to the introduction. With this we tried to go more in depth and explain why no biomarker-based test is currently being used in clinical practice. Additionally, more background was added to the introduction adding more basic information on the mutations which are used within our test.
With this letter we hope to have adequately addressed your comments and hope to have answered your questions. We would again like to thank you for your time and effort and look forward to hearing from you.
With kind regards,
Caroline Sieverink
Reviewer 2 Report
The study titled "Clinical Validation of a Urine Test (Uromonitor-V2®) for the Surveillance of Non-Muscle Invasive Bladder Cancer Patients", aims to clinically validate the efficiency of Uromonitor-V2 for the surveillance of NMIBC patients. The study is promising in terms of its high NPV, non-invasive nature of testing and advantages such as a short test to results time and easy to use/implement methodology such a qPCR. However, unfortunately, the manuscript suffers from poor English language throughout and fails to provide sufficient background information on the study topic, graphical representation of at least some of the results, and justification for the selection and addition of KRAS gene to the previously published (in Frontier in Genetics) Uromonitor panel of TERT and FGFR3 genes in additional to the following concerns listed below:
- The introduction section lacks important information such as a brief elaboration of the current urinary biomarkers and the reason why biomarker based urine tests that have shown promising results and high NPV (98-99%) but still could not be implemented into clinical practice.
- Also neither the criteria behind using a specific set of genes for this new test nor the unique strength or advantage of using this specific set of genes in the new urine test has been explained in the manuscript.
- The manuscript lacks figures, such as, for example, the workflow of the Uromonitor-V2 methodology/test, graphs (instead of only tables) representing the percentages in results.
- Patient information is repeated in results while it has already been discussed in methods
- Moreover, this study lacks novelty, as the Uromonitor V2 seems like a slightly modified version of Uromonitor (with the addition of KRAS gene to the panel) already published by the same research group earlier in Frontiers in genetics
- The size the study cohort is very small, which makes the results less reliable.
- The conclusion section states that "The test displayed its potential as an alternative to the current follow-up methods in two studies on independent cohorts, showing comparable promising results [12].", but there is a literature cited only for one study and not two. This discrepancy needs to be clarified/rectified.
Author Response
To the reviewer,
We would also like to thank you for taking the time and effort to review our manuscript, your feedback is much appreciated. We would like to use this letter to address your comments and hope that by implementing the feedback, we have improved our manuscript.
Point: The study titled "Clinical Validation of a Urine Test (Uromonitor-V2®) for the Surveillance of Non-Muscle Invasive Bladder Cancer Patients", aims to clinically validate the efficiency of Uromonitor-V2 for the surveillance of NMIBC patients. The study is promising in terms of its high NPV, non-invasive nature of testing and advantages such as a short test to results time and easy to use/implement methodology such a qPCR. However, unfortunately, the manuscript suffers from poor English language throughout and fails to provide sufficient background information on the study topic, graphical representation of at least some of the results, and justification for the selection and addition of KRAS gene to the previously published (in Frontier in Genetics) Uromonitor panel of TERT and FGFR3 genes in additional to the following concerns listed below:
Response: Thank you for your feedback and your suggestions. Please be informed that the manuscript underwent an extensive check and editing of English language. For our additional response we would like to refer you to the text below.
Point 1: The introduction section lacks important information such as a brief elaboration of the current urinary biomarkers and the reason why biomarker based urine tests that have shown promising results and high NPV (98-99%) but still could not be implemented into clinical practice.
Response: Background information regarding the current (commercially) available biomarkers was added to the introduction section, explaining more in depth why, although showing promising results at first, none of the current biomarkers are used in clinical practice. This is mostly due to the fact that the older biomarkers lack adequate NPV and sensitivity. Over the last years a number of biomarker-based urine tests with significantly higher (and clinically relevant) NPV’s have emerged, but none of these markers are routinely used since they need clinical validation before implementation.
Point 2: Also neither the criteria behind using a specific set of genes for this new test nor the unique strength or advantage of using this specific set of genes in the new urine test has been explained in the manuscript.
Response: More background on this was added to the introduction section of our manuscript. The previous version of the urine test (the Uromonitor-V1) only tested for TERTp and FGFR3 mutations. With the second version (Uromonitor-V2) the testing for KRAS mutations was added, since bladder tumors were left undetected with the Uromonitor-V1. Mutations in RAS genes are found in 13% of all bladder cancer patients. Moreover, KRAS mutations and FGFR3 mutations are known to be mutually exclusive events in bladder cancer. It was therefore hypothesized that adding KRAS to the test could increase the detecting capability of the Uromonitor test up to clinically relevant levels, as was already found in a small sub-analysis of the Uromonitor-V1.
Point 3: The manuscript lacks figures, such as, for example, the workflow of the Uromonitor-V2 methodology/test, graphs (instead of only tables) representing the percentages in results.
Response: Please find more figures and graphs added to our manuscript. Figure 1 shows the ROC curves of urine cytology and the Uromonitor-V2 test on patients with a history of NMIBC, making the sensitivity and specificity of our test compared to cytology more visual. Figure 2 shows a visualization of the workflow.
Point 4: Patient information is repeated in results while it has already been discussed in methods
Response: We would like to thank the reviewer for this comment, but would like to ask for a clarification, since it was not clear which patient information is repeated. In the methods section the patient information explains the make-up of the three patient groups and the in- and exclusion criteria per group. In the results section the number of patients per group are explained, while also comparing the three groups based on their characteristics.
Point 5: Moreover, this study lacks novelty, as the Uromonitor V2 seems like a slightly modified version of Uromonitor (with the addition of KRAS gene to the panel) already published by the same research group earlier in Frontiers in genetics
Response: The publication in Frontiers in Genetics uses the previous version of our test, the Uromonitor-V1, which tested for TERTp and FGFR3 mutations. Within this study a small subset of bladder cancer patients was also tested for KRAS mutations. Twenty-four patients in total were tested using all three markers. This small sub-analysis showed such promising results (with a sensitivity, specificity and NPV of 100%, 83.8% and 100%, respectively) that it was decided to validate these results in a larger cohort of NMIBC patients, which was the aim of the current study.
Point 6: The size the study cohort is very small, which makes the results less reliable.
Response: We acknowledge this to be true. Ideally, a larger number of patients would have been enrolled in our study to make for more reliable results. However, we believe that the current results are promising and that future research in a larger, phase III multicenter trial should be done to validate these results.
Point 7: The conclusion section states that "The test displayed its potential as an alternative to the current follow-up methods in two studies on independent cohorts, showing comparable promising results [12].", but there is a literature cited only for one study and not two. This discrepancy needs to be clarified/rectified.
Response: Please find a clarification for this statement in the conclusions section. It is now explained that the current validation study and the previous study on the Uromonitor show comparable findings.
With this letter we hope to have adequately addressed your comments and hope to have answered your questions. Once more, we would like to thank you for your time and effort and look forward to hearing from you.
With kind regards,
Caroline Sieverink
Round 2
Reviewer 2 Report
To the Authors,
//Point 4: Patient information is repeated in results while it has already been discussed in methods
Response: We would like to thank the reviewer for this comment, but would like to ask for a clarification, since it was not clear which patient information is repeated. In the methods section the patient information explains the make-up of the three patient groups and the in- and exclusion criteria per group. In the results section the number of patients per group are explained, while also comparing the three groups based on their characteristics.//
I meant that the title patients is repeated both in methods and results section which is misleading. Now, I understand the reasoning behind the placement of patients' information both in results and methods. I would like to recommend changing the patients section title in results section to "Patients' characteristics" which will remove ambiguity.
Overall, the manuscript has been significantly improved with the addition of requested information and necessary figures based on the comments. Now, the article is suitable for publication in Diagnostics. Congratulations!
This manuscript is a resubmission of an earlier submission. The following is a list of the peer review reports and author responses from that submission.